# Fairness-aware Missing Data Imputation

**Yiliang Zhang**
University of Pennsylvania
Philadelphia, PA 19104, USA
zylthu14@sas.upenn.edu

**Qi Long**
University of Pennsylvania
Philadelphia, PA 19104, USA
qlong@upenn.edu

## Abstract

Missing values are ubiquitous in real-world datasets and are known to cause unfairness in a machine learning algorithm's decision-making process. However, there has been limited work that aims to mitigate the unfairness associated with missing data imputation. In this paper, we first derive a positive information-theoretic lower bound for the imputation fairness when using ground-truth conditional distribution for missing data imputation. Furthermore, we propose a novel missing data imputation model, known as fairness-aware imputation GAN (FIGAN), which provides accurate imputations while achieving imputation fairness. Through experiments, we illustrate that FIGAN can significantly improve imputation fairness, compared to the existing imputation methods. At the same time, FIGAN can also achieve competitive imputation accuracy.

## 1 Introduction

Missing values frequently occur in real-world datasets. While it has been well known that the absence of values will harm the performance of downstream predictions, recent works have connected the missing data with discrimination of decisions made by the downstream machine learning algorithms [Martínez-Plumed et al., 2019, Wang and Singh, 2021, Jeong et al., 2022, Zhang and Long, 2021b].

Investigations on the discrimination of algorithm's decisions, namely algorithmic fairness, have been at the center of the trustworthiness of artificial intelligence. Despite that there have been many works that aim to improve the fairness of a prediction model, there have been limited solutions that address the fairness concerns raised by the missing data. In the presence of missing data, two classical solutions are: (1) drop the samples that contain missing values. (2) impute the missing values. To the best of our knowledge, there has not been any work that aims to improve fairness when one adopts an imputation model to handle the missing values. In addition, there has not been any work that provides theoretical analysis on the fairness of imputation models.

In this paper, we first provide a theoretical analysis of imputation models that are correctly specified, in terms of the imputation fairness risk [Zhang and Long, 2021b]. In a more general setting, we propose fairness-aware imputation GAN (FIGAN), a GAN-based imputation model that can effectively control imputation unfairness. Through experiments, we illustrate the outstanding performance of FIGAN on both synthetic and real datasets.

### 1.1 Related works

Algorithmic fairness focuses on the existence of discrimination during the decision-making process of a machine learning algorithm. The existing work can be divided into group fairness [Feldman et al., 2015, Kamiran and Calders, 2009, Kamishima et al., 2011] and individual fairness [Dwork et al., 2012]. Group fairness emphasizes that members from different sensitive groups should be treated similarly by the algorithm, while individual fairness emphasizes that similar individuals should be

2022 Trustworthy and Socially Responsible Machine Learning (TSRML 2022) co-located with NeurIPS 2022.

treated similarly. In this work, we focus on the group fairness of the missing data imputation methods. We adopt the imputation fairness defined in Zhang and Long [2021b] as our fairness notion.

Recently, a line of research studied the relationship between missing data and algorithmic fairness. Martínez-Plumed et al. [2019] analyzed different causes of missing values, and empirically discovered that missing values may harm the fairness of the downstream prediction tasks. Following this idea, Wang and Singh [2021] conducted more extensive empirical evaluations and demonstrated that applying reweighting techniques can help mitigate the unfairness of the downstream tasks. Goel et al. [2021] established a causal framework to analyze the effect of missing data on the fairness of downstream tasks. Later on, Jeong et al. [2022] proposes a tree-based method for fair prediction in the presence of missing data, which does not require any missing data imputation. Zhang and Long [2021a] provided the first known theoretical results on fairness guarantee in the analysis of incomplete data. However, as arguably the most popular approach for handling missing data, missing data imputation has not been well-studied in terms of algorithmic fairness. To the best of our knowledge, the only work that studies the fairness of imputation methods is Zhang and Long [2021b], in which the authors assessed the fairness properties of popular imputation models through extensive numerical experiments. In this paper, we are among the first to consider the problem of improving the fairness of missing data imputation and propose a novel imputation method that can yield accurate imputation while guaranteeing imputation fairness.

There is a large body of literature on missing data imputation. Classical imputation approaches include using mean values of observed data, chained equations [Van Buuren and Oudshoorn, 1999], matrix completion [Mazumder et al., 2010] and EM algorithm [Dempster et al., 1977]. With the recent development of machine learning, tree-based methods [Stekhoven and Bühlmann, 2012], optimal transport [Muzellec et al., 2020] and deep-learning-based methods [Yoon et al., 2018, Dai et al., 2021, Li et al., 2019, Bansal et al., 2021, Tashiro et al., 2021, You et al., 2020] are adopted for imputations. Our proposed imputation model is based on the Generative Adversarial Networks (GAN) [Goodfellow et al., 2020]. While there has been recent work on developing GAN models for fair data generation [Xu et al., 2018, 2019b,a], to the best of our knowledge, our work proposes the first known fairness-aware imputation model that leverages the power of GAN.

## 2 Preliminaries

### 2.1 Problem setup

We consider the problem of imputing missing values in dataset $\mathbf{X} = \{\mathbf{X}_i\}_{i=1}^n \in \mathbb{R}^{n \times p}$. Each sample $\mathbf{X}_i = \{\mathbf{X}_{i1}, \ldots, \mathbf{X}_{ip}\}$ is associated with a sensitive attribute $A = \{A_i\}_{i=1}^n$ (Figure 1). Without loss of generality, we assume that there are two sensitive groups, that is, $A_i \in \{0, 1\}$. We divide the features into two groups: $\mathbf{X} = \mathbf{X}^{miss} \cup \mathbf{X}^{obs}$ (Figure 1), where $\mathbf{X}^{miss} = (\mathbf{X}_{ij}^{miss})$ denotes the subset of features that have missing values, and $\mathbf{X}^{obs} = (\mathbf{X}_{ij}^{obs})$ denotes the subset of features that are fully observed. We define a missing data indicator $M = (m_{ij}) \in \{0, 1\}^{n \times p}$ where $m_{ij} = \mathbf{1}\{\mathbf{X}_{ij} \text{ is missing}\}$. In addition, we define $M_i = \mathbf{1}\{\mathbf{X}_i \text{ has missing value}\}$ (Figure 1). The missing data patterns can be classified into three mechanisms: *missing completely at random* (MCAR), *missing at random* (MAR) and *missing not at random* (MNAR) [Little and Rubin, 2019]. Data are said to be MCAR if the missingness is independent of both observed and missing values, i.e., $M \perp \mathbf{X}$; Data are said to be MAR if the missingness only depends on observed values; Data that are not MCAR or MAR are said to be MNAR, i.e., $M \not\perp \mathbf{X}^{miss}$. We further denote the imputation model of interest by $\hat{\mathbf{X}} = f_{imp}(\mathbf{X}^{obs}, A)$. A popular and useful metric that evaluates the performance of an imputation model is the *mean squared imputation error* (MSIE):

$$\text{MSIE}(f_{imp}) = \frac{\sum_{(i,j)} \left(\hat{\mathbf{X}}_{ij}(f_{imp}) - \mathbf{X}_{ij}\right)^2 m_{ij}}{\sum_{(i,j)} m_{ij}} \tag{1}$$

To assess the fairness of an imputation model, we define the MSIE within each sensitive group: $\text{MSIE}_{A=a}(f_{imp}) = \sum_{(i,j)}(\hat{\mathbf{X}}_{ij}^a(f_{imp}) - \mathbf{X}_{ij}^a)^2 m_{ij}^a / \sum_{(i,j)} m_{ij}^a$ where $\mathbf{X}^a = (\mathbf{X}_{ij}^a)$ denote the samples within sensitive group $A = a$. Equipped with these definitions, we formally introduce the fairness notion for the imputation process.

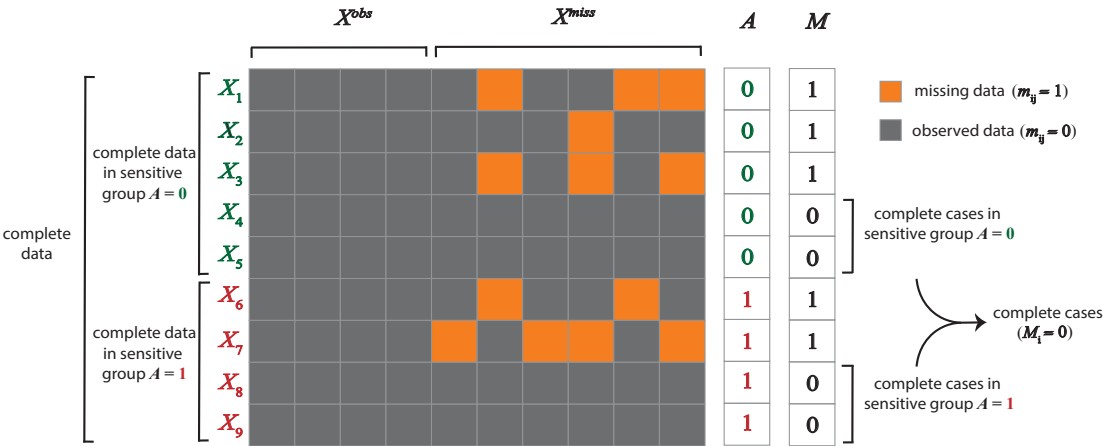

Figure 1: Illustration of notations

**Definition 2.1** (Imputation Fairness). *For a given imputation model $f_{imp}$, the imputation fairness risk (IFR) is defined as*

$$IFR(f_{imp}) = |MSIE_{A=0}(f_{imp}) - MSIE_{A=1}(f_{imp})| \qquad (2)$$

Intuitively, the imputation fairness risk (IFR) measures the difference in imputation error between groups defined by a sensitive attribute. A perfectly fair imputation should provide equally accurate imputations for samples from different groups defined by a sensitive attribute.

## 2.2 A lower bound for IFR: when true imputation model is used

Conceptually, an imputation model learns the distribution of missing data $\mathbf{X}^{miss}$, given the observed information $\mathbf{X}^{obs}$ and $A$. That is, the imputation model $f_{imp}(\mathbf{X}^{obs}, A)$ approximates the conditional distribution $\mathbb{P}(\mathbf{X}^{miss}|\mathbf{X}^{obs}, A)$. While the true conditional distribution $\mathbb{P}(\mathbf{X}^{miss}|\mathbf{X}^{obs}, A)$ seems to be an ideal imputation, we argue that there can be inherent imputation fairness risk associated. Indeed, in the following theorem, we provide an information-theoretic lower bound of the IFR for single-pattern missing data (i.e., for each sample, either it is fully observed, or the entire row in $\mathbf{X}^{miss}$ will be missing simultaneously), when using the true conditional distribution $\mathbb{P}(\mathbf{X}^{miss}|\mathbf{X}^{obs}, A)$ for imputation.

**Theorem 1.** *For a dataset that has single-pattern missing data (i.e., $m_{ij}^{miss} \equiv M_i$), assume the true conditional distribution $\mathbb{P}(\mathbf{X}^{miss}|\mathbf{X}^{obs}, A)$ is adopted for imputation, then imputation fairness risk has the lower bound:*

$$\mathbb{E}[IFR(f_{imp})] \geq \frac{2}{K} \left| \sum_{j=1}^{K} \left( \mathbb{E}_{\mathbf{X}^{obs}|M=1,A=0}\sigma_{0,j}^2 - \mathbb{E}_{\mathbf{X}^{obs}|M=1,A=1}\sigma_{1,j}^2 \right) \right|$$

*where $K$ is the number of features that has missing values (i.e., the number of columns in $\mathbf{X}^{miss}$), and*

$$\sigma_{a,j}^2 = Var(\mathbf{X}_{ij}^{miss}|M_i = 1, A_i = a, \mathbf{X}^{obs})$$

*is the conditional variance of the missing data in the $j$-th column of $\mathbf{X}^{miss}$.*

The proof of the theorem is provided in Appendix A.1. Our results imply that even if the ground truth conditional distribution is used for imputation, there exists a positive information-theoretic lower bound for the imputation fairness risk. In practice, the true conditional distribution is not accessible. However, it is possible to correctly specify the conditional distribution model (e.g., gaussian, uniform, etc.) and construct an unbiased estimate of the distribution. Consider a general case where the true conditional distribution is parameterized by $\mathbb{P}(\mathbf{X}^{miss}|\mathbf{X}^{obs}, A) = f_{imp}(\theta, \mathbf{X}^{obs}, A)$. We consider

the family of imputation models that are correctly-specified: $\{f_{imp}(\hat{\theta}, \mathbf{X}^{obs}, A)\}_{\hat{\theta} \in \Theta}$. Similar to Theorem 1, one can derive information lower bound for these correctly-specified imputation models. The lower bound will not only include the conditional variance $Var(\mathbf{X}_{ij}^{miss}|M_i = 1, A_i = a, \mathbf{X}^{obs})$, but also the bias of the parameter estimate $\hat{\theta}$. A detailed analysis is provided in the Appendix A.1.

## 3  Fairness-awared Imputation GAN (FIGAN)

In Section 2, we have provided discussions on the imputation fairness risk of those correctly-specified imputation models. In general, however, one may not have much information to construct a correctly-specified parametric imputation model. As an alternative solution, state-of-the-art deep learning models are adopted to construct non-parametric imputation models. Following this idea, in this paper, we propose a fairness-aware imputation model based on generative adversarial networks (GAN). The original GAN [Goodfellow et al., 2020] learns the underlying distribution of a given dataset and artificially generates fake data that are similar to the original data.

We proposed Fairness-aware Imputation GAN (FIGAN) in Algorithm 1, which consists of two neural networks: generator $\mathcal{G}$ and discriminator $\mathcal{D}$. Figure 2 shows the FIGAN's framework. At a high level, the generator $\mathcal{G}$ uses observed information of a sample, $\mathbf{X}^{obs}$, sensitive attribute $A$ and random noise $z$ to impute the missing values $\mathbf{X}^{miss}$. The discriminator $\mathcal{D}$ learns to discriminate the imputed data $\hat{\mathbf{X}}^{miss}$ from the ground truth $\mathbf{X}^{miss}$. To control the imputation fairness risk, a fairness regularization is imposed into the training objective function. For the discriminator $\mathcal{D}$, it is trained by minimizing the loss function

$$\mathcal{L}_{\mathcal{D}} = \mathbb{E}[\log(\mathcal{D}(\mathbf{X}^{miss}))] + \mathbb{E}[\log(1 - \mathcal{D}(\mathcal{G}(\mathbf{X}^{obs}, A, z)))] \tag{3}$$

The generator $\mathcal{G}$ is trained by minimizing the loss function

$$\mathcal{L}_{\mathcal{G}} = \mathbb{E}[\log(\mathcal{D}(\mathcal{G}(\mathbf{X}^{obs}, A, z)))] + \lambda_{acc}\text{MSIE}(\mathcal{G}) + \lambda_{fair}\text{IFR}(\mathcal{G}) \tag{4}$$

in which $\lambda_{acc}$ and $\lambda_{fair}$ are tunable hyper-parameters. The first term in $\mathcal{L}_{\mathcal{G}}$ forces the generator $\mathcal{G}$ to produce imputed values that can "fool" the discriminator $\mathcal{D}$, and the second term forces to produce accurate imputations, the third term regularizes the model to have small imputation fairness risk.

---

**Algorithm 1** `Training Fairness-aware Imputation GAN (FIGAN)`

---

**Input:** Complete cases (see definition in Figure 1) $\mathbf{X}_{cc} = \mathbf{X}_{cc}^{obs} \cup \mathbf{X}_{cc}^{miss}$, sensitive attribute $A_{cc}$, number of iteration EPOCHS.
**Output:** Fairness-aware Imputation GAN (i.e., $\mathcal{G}$ and $\mathcal{D}$).

   **for** $t$ in $\{1, \ldots, \text{EPOCHS}\}$ **do**
      (1) Sample the noise $z \sim \mathcal{N}(0, \mathbf{I})$.

      (2) Impute the missing values via $\hat{\mathbf{X}}_{cc}^{miss} = \mathcal{G}(\mathbf{X}_{cc}^{obs}, A_{cc}, z)$.

      (3) Forward the imputed values $\hat{\mathbf{X}}_{cc}^{miss}$ and true missing values $\mathbf{X}_{cc}^{miss}$ into discriminator $\mathcal{D}$.

      (4) Update the discriminator using $\mathcal{L}_{\mathcal{D}}$ (3): $\theta_{\mathcal{D}}^{(t+1)} \leftarrow Adam(\theta_{\mathcal{D}}^{(t)}, \nabla\mathcal{L}_{\mathcal{D}})$

      (5) Update the generator using $\mathcal{L}_{\mathcal{G}}$ (4): $\theta_{\mathcal{G}}^{(t+1)} \leftarrow Adam(\theta_{\mathcal{G}}^{(t)}, \nabla\mathcal{L}_{\mathcal{G}})$
   **end for**

---

Noticed that the loss function $\mathcal{L}_{\mathcal{D}}$ contains the ground truth value $\mathbf{X}^{miss}$, FIGAN will be trained on complete cases (Figure 1) in which the ground truth values $\mathbf{X}^{miss}$ are known. Specifically, for each complete case, we can generate the imputed values using fully observed features $\mathbf{X}^{obs}$ and train FIGAN using $\mathbf{X}^{miss}$ and $\mathcal{G}(\mathbf{X}^{obs}, A, z)$.

## 4  Empirical Results

In this section, we study the imputation performance of proposed FIGAN and 5 other existing methods on both synthetic and real datasets: MICE [Van Buuren and Oudshoorn, 1999], Misforest [Stekhoven

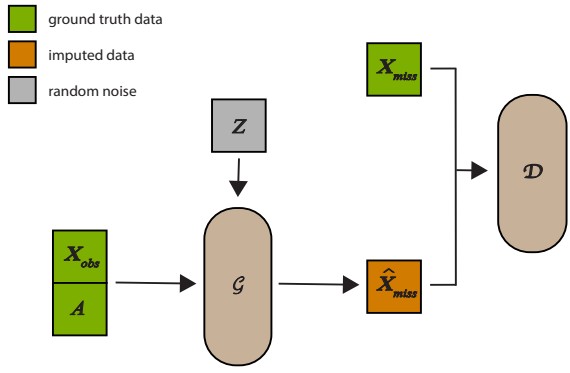

Figure 2: FIGAN framework

and Bühlmann, 2012], SoftImpute [Mazumder et al., 2010], Sinkhorn [Muzellec et al., 2020] and Gain [Yoon et al., 2018]. We illustrate that FIGAN is capable of achieving decent imputation accuracy while providing fairer imputation results.

## 4.1 Synthetic datasets

We generate the synthetic dataset via a time-series model. In particular, $X_{i+1} = 0.1X_i + \epsilon_{i+1}$ with $\epsilon_i \sim \mathcal{N}(0,1)$ being independent white noise. We set the sample size $n = 1000$, number of features $p = 25$. We artificially generate missing values for 4 features, under three missing mechanisms: logit$(\mathbb{P}(M_i = 1)) = 0.7$; MAR, logit$(\mathbb{P}(M_i = 1)) = 0.1\mathbb{E}_{empirical}\mathbf{X}^{obs}$; MNAR, logit$(\mathbb{P}(M_i = 1)) = 0.1\mathbb{E}_{empirical}\mathbf{X}^{miss}$. The results are summarized into Table 1.

| Imputation model | | MICE | Missforest | SoftImpute | OptimalTransport | Gain | FIGAN (ours) |
|---|---|---|---|---|---|---|---|
| MCAR | MSIE | 0.440 | 0.244 | 0.220 | 0.240 | 0.506 | **0.163** |
| | Fairness | 0.017 | 0.012 | 0.011 | 0.010 | 0.195 | **0.006** |
| MAR | MSIE | 0.439 | 0.244 | 0.220 | 0.240 | 0.504 | **0.157** |
| | Fairness | 0.015 | 0.012 | 0.011 | 0.011 | 0.191 | **0.007** |
| MNAR | MSIE | 0.440 | 0.243 | 0.220 | 0.240 | 0.507 | **0.165** |
| | Fairness | 0.016 | 0.010 | 0.010 | 0.010 | 0.195 | **0.009** |

Table 1: MSIE and fairness risk of different imputation models on synthetic datasets. The mean values over 50 repeats are reported.

The results in Table 1 show that FIGAN has the smallest imputation fairness and smallest MSIE simultaneously. This outperformance demonstrates FIGAN's potential to provide accurate and fair imputations in practice.

## 4.2 Real datasets

To study the performance of FIGAN in practice, we conduct experiments on two real-world datasets, COMPAS [Northpointe, 2010] and ADNI. In both experiments, to assess the MSIE, we will artificially generate missing values, and compare the imputed values with the ground truth values. More details of the datasets and experiment setup can be found in the Appendix A.2.

**COMPAS recidivism dataset:**

Correctional Offender Management Profiling for Alternative Sanctions (COMPAS) [Northpointe, 2010] dataset contains records of defendants from Broward County from 2013 and 2014. In our analysis, gender is treated as the sensitive attribute, and we only extract and use 12 non-categorical features in our experiments. We generate missing values under three missing mechanisms: MCAR,

logit($\mathbb{P}(M_i = 1)$) = 0.7; MAR, logit($\mathbb{P}(M_i = 1)$) = $3+0.4\mathbb{E}_{empirical}\mathbf{X}^{obs}$; MNAR, logit($\mathbb{P}(M_i = 1)$) = $-2\mathbb{E}_{empirical}\mathbf{X}^{miss}$. The results are summarized in Table 2. From the table, we observe that FIGAN consistently has significantly lower imputation fairness risks, compared to all the other imputation methods. In addition, FIGAN also achieves comparable, if not the best, MSIE in the experiment.

| | Imputation model | | MICE | Missforest | SoftImpute | OptimalTransport | Gain | FIGAN (ours) |
|---|---|---|---|---|---|---|---|---|
| COMPAS | MCAR | MSIE | 0.778 | 0.908 | 0.748 | 0.861 | 1.261 | **0.734** |
| | | Fairness | 0.069 | 0.186 | 0.076 | 0.065 | 0.097 | **0.022** |
| | MAR | MSIE | 0.799 | 0.938 | 0.895 | 0.891 | 1.267 | **0.752** |
| | | Fairness | 0.087 | 0.192 | 0.098 | 0.100 | 0.089 | **0.018** |
| | MNAR | MSIE | 0.850 | 1.318 | **0.639** | 1.429 | 1.457 | 0.810 |
| | | Fairness | 0.071 | 0.045 | 0.073 | 0.122 | 0.091 | **0.015** |
| ADNI | MCAR | MSIE | **0.254** | 0.774 | 0.432 | 1.039 | 1.488 | 0.390 |
| | | Fairness | 0.172 | 0.226 | 0.172 | 0.166 | 0.437 | **0.053** |
| | MAR | MSIE | 0.248 | 0.808 | 0.430 | 1.047 | 1.582 | **0.150** |
| | | Fairness | 0.137 | 0.224 | 0.181 | 0.132 | 0.289 | **0.049** |
| | MNAR | MSIE | **0.252** | 0.989 | 0.432 | 1.022 | 1.420 | 0.259 |
| | | Fairness | 0.101 | 0.231 | 0.204 | 0.201 | 0.257 | **0.053** |

Table 2: MSIE and fairness risk of different imputation models on COMPAS and ADNI datasets. The mean values over 50 repeats are reported.

**ADNI dataset:**

Launched in 2003, the primary goal of the Alzheimer's Disease Neuroimaging Initiative (ADNI) database is to test whether it is possible to measure the progression of mild cognitive impairment and early Alzheimer's disease. The database contains patients' positron emission tomography (PET), serial magnetic resonance imaging (MRI), other biological markers, and clinical and neuropsychological assessments. The dataset adopted in this paper contains genomic information of 649 patients, who potentially have Alzheimer's disease. Among the 19k features in the original dataset, we only choose the first 1000 columns from the raw data in experiments. We set race as the sensitive attribute and artificially generate missing values for the first 10 columns in the dataset. The missing data are generated under three missing mechanisms: logit($\mathbb{P}(M_i = 1)$) = 0.5; MAR, logit($\mathbb{P}(M_i = 1)$) = $0.1\mathbb{E}_{empirical}\mathbf{X}^{obs}$; MNAR, logit($\mathbb{P}(M_i = 1)$) = $0.1\mathbb{E}_{empirical}\mathbf{X}^{miss}$. The results are summarized in Table 2. Similar to the experiments on the COMPAS dataset, from Table 2, we also observe that FIGAN consistently has significantly lower imputation fairness risk and achieves comparable MSIE in the experiment.

## 5 Discussions

In this paper, we study the fairness associated with missing data imputation. We first provide a theoretical analysis that suggests that there exists inherent imputation unfairness when using the ground truth conditional distribution of the missing data for imputation. Furthermore, we propose FIGAN, a GAN-based imputation model that empirically provides accurate and fair imputations.

We believe there is much room remaining for future study. Firstly, there are widely used fairness notions based on causality, and investigations based on these causal fairness notions could provide unique insights as well. Secondly, other than GAN, there have been other generative models such as autoencoders and diffusion models that achieve decent performance in synthetic data generation. We believe that fair imputation models based on these generative models can be an exciting direction for future study. Thirdly, theoretical analysis on the fairness of FIGAN could also be insightful, we expect it to help better understand the intuitions and strategies of improving fairness during missing data imputation.

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

# A Appendix

## A.1 Theoretical analysis on correctly-specified imputation model

We begin by providing the proof of Theorem 1.

*Proof.* Taking the expectation of the mean squared imputation error, we have that

$$\mathbb{E}[\text{MSIE}_{A=a}(f_{imp})] = \mathbb{E}\left[\frac{\sum_{(i,j)}(\hat{\mathbf{X}}_{ij}^a - \mathbf{X}_{ij}^a)^2 m_{ij}^a}{\sum_{(i,j)} m_{ij}^a}\right] = \frac{1}{K}\sum_{j=1}^{K}\mathbb{E}((\hat{\mathbf{X}}_{ij}^{miss} - \mathbf{X}_{ij}^{miss})^2|M_i = 1, A_i = a)$$

where the second equation holds since all the features in $\mathbf{X}^{miss}$ are missing simultaneously. By law of total expectations,

$$\mathbb{E}[\text{MSIE}_{A=a}(f_{imp})] = \frac{1}{K}\sum_{j=1}^{K}\mathbb{E}_{\mathbf{X}^{obs}|M=1,A=a}\mathbb{E}((\hat{\mathbf{X}}_{ij}^{miss} - \mathbf{X}_{ij}^{miss})^2|M_i = 1, A_i = a, \mathbf{X}^{obs})$$
(5)

Since we only have the information about $(\mathbf{X}^{obs}, A_i, M_i)$ during the imputation, the imputed value $\hat{\mathbf{X}}_{ij}^{miss}$ is independent of the true value $\mathbf{X}_{ij}^{miss}$, conditioned on $(\mathbf{X}^{obs}, A_i, M_i)$. Moreover, since we are using the true conditional distribution $\mathbb{P}(\mathbf{X}^{miss}|\mathbf{X}^{obs}, A)$, $\hat{\mathbf{X}}_{ij}^{miss}$ has the same conditional distribution as $\mathbf{X}_{ij}^{miss}$. Hence we have that

$$\mathbb{E}_{\mathbf{X}^{obs}}\mathbb{E}((\hat{\mathbf{X}}_{ij}^{miss} - \mathbf{X}_{ij}^{miss})^2|M_i = 1, A_i = a, \mathbf{X}^{obs}) = 2\mathbb{E}_{\mathbf{X}^{obs}|M=1,A=a}\sigma_{a,j}^2$$
(6)

with $\sigma_{a,j}^2 = Var(\mathbf{X}_{ij}^{miss}|M_i = 1, A_i = a, \mathbf{X}^{obs})$. Finally, by triangle inequality,

$$\mathbb{E}[\text{IFR}(f_{imp})] = \mathbb{E}|\text{MSIE}_{A=0}(f_{imp}) - \text{MSIE}_{A=1}(f_{imp})| \geq |\mathbb{E}[\text{MSIE}_{A=0}(f_{imp})] - \mathbb{E}[\text{MSIE}_{A=1}(f_{imp})]|$$

combining equation (5) and (6) into the above expression completes the proof. $\square$

Moving one step further, we provide an analysis of the imputation fairness for correctly specified imputation models.

**Theorem 2.** *For a dataset that has single-pattern missing data (i.e., $m_{ij}^{miss} \equiv M_i$), assume a correctly specified imputation model $f_{imp}(\hat{\theta}, \mathbf{X}^{obs}, A)$ is adopted for imputation, then imputation fairness risk has the lower bound:*

$$\mathbb{E}[IFR(f_{imp})] \geq \frac{2}{K}\left|\sum_{j=1}^{K}\left(\mathbb{E}_{\mathbf{X}^{obs}|M=1,A=0}\tilde{\sigma}_{0,j}^2 - \mathbb{E}_{\mathbf{X}^{obs}|M=1,A=1}\tilde{\sigma}_{1,j}^2\right)\right|$$

*where $K$ is the number of features that has missing values (i.e., the number of columns in $\mathbf{X}^{miss}$), and*

$$\tilde{\sigma}_{a,j}^2 = Var(\mathbf{X}_{ij}^{miss}|M_i = 1, A_i = a, \mathbf{X}^{obs}) + (\hat{\theta} - \theta)\mathbb{E}(\mathbf{X}_{ij}^{miss}(\mathbf{X}_{ij}^{miss} - 2\mathbb{E}(\mathbf{X}_{ij}^{miss}|M_i = 1, A_i = a, \mathbf{X}^{obs}))$$

$$* \frac{\partial f_{imp}(\theta, M_i = 1, A_i = 1)/\partial\theta|_{\theta=\hat{\theta}}}{f_{imp}(\theta, M_i = 1, A_i = 1)}|M_i = 1, A_i = a, \mathbf{X}^{obs}) + o(\hat{\theta} - \theta)$$

*For consistent estimate $\hat{\theta}$, we have that $\tilde{\sigma}_{a,j}^2 \to \sigma_{a,j}^2$ as $n \to \infty$.*

*Proof.* From the proof of Theorem 1, we know that

$$\mathbb{E}[\text{MSIE}_{A=a}(f_{imp})] = \frac{1}{K}\sum_{j=1}^{K}\mathbb{E}_{\mathbf{X}^{obs}|M=1,A=a}\mathbb{E}((\hat{\mathbf{X}}_{ij}^{miss} - \mathbf{X}_{ij}^{miss})^2|M_i = 1, A_i = a, \mathbf{X}^{obs})$$

In addition, the imputed value $\hat{\mathbf{X}}_{ij}^{miss}$ is independent of the true value $\mathbf{X}_{ij}^{miss}$, conditioned on $(\mathbf{X}^{obs}, A_i, M_i)$. We have that

$$\mathbb{E}_{\mathbf{X}^{obs}}\mathbb{E}((\hat{\mathbf{X}}_{ij}^{miss} - \mathbf{X}_{ij}^{miss})^2|M_i = 1, A_i = a, \mathbf{X}^{obs})$$

$$=\mathbb{E}_{\mathbf{X}^{obs}}[\mathbb{E}((\hat{\mathbf{X}}_{ij}^{miss})^2|M_i = 1, A_i = a, \mathbf{X}^{obs}) + \mathbb{E}((\mathbf{X}_{ij}^{miss})^2|M_i = 1, A_i = a, \mathbf{X}^{obs})$$

$$-2\mathbb{E}(\hat{\mathbf{X}}_{ij}^{miss}|M_i = 1, A_i = a, \mathbf{X}^{obs})\mathbb{E}(\mathbf{X}_{ij}^{miss}|M_i = 1, A_i = a, \mathbf{X}^{obs})]$$

For correctly-specified imputation model $f_{imp}(\hat{\theta}, \mathbf{X}^{obs}, A)$, applying the Taylor expansion yields:

$$\mathbb{E}(\hat{\mathbf{X}}_{ij}^{miss}|M_i = 1, A_i = a, \mathbf{X}^{obs}) = \mathbb{E}(\mathbf{X}_{ij}^{miss}|M_i = 1, A_i = a, \mathbf{X}^{obs})$$

$$+(\hat{\theta} - \theta)\mathbb{E}(\mathbf{X}_{ij}^{miss}\frac{\partial f_{imp}(\theta, M_i = 1, A_i = 1)/\partial\theta|_{\theta=\hat{\theta}}}{f_{imp}(\theta, M_i = 1, A_i = 1)}|M_i = 1, A_i = a, \mathbf{X}^{obs}) + o(\hat{\theta} - \theta)$$

and

$$\mathbb{E}((\hat{\mathbf{X}}_{ij}^{miss})^2|M_i = 1, A_i = a, \mathbf{X}^{obs}) = \mathbb{E}((\mathbf{X}_{ij}^{miss})^2|M_i = 1, A_i = a, \mathbf{X}^{obs})$$

$$+(\hat{\theta} - \theta)\mathbb{E}((\mathbf{X}_{ij}^{miss})^2\frac{\partial f_{imp}(\theta, M_i = 1, A_i = 1)/\partial\theta|_{\theta=\hat{\theta}}}{f_{imp}(\theta, M_i = 1, A_i = 1)}|M_i = 1, A_i = a, \mathbf{X}^{obs}) + o(\hat{\theta} - \theta)$$

combine together above three equations, we have that

$$\mathbb{E}_{\mathbf{X}^{obs}}\mathbb{E}((\hat{\mathbf{X}}_{ij}^{miss} - \mathbf{X}_{ij}^{miss})^2|M_i = 1, A_i = a, \mathbf{X}^{obs}) = 2\mathbb{E}_{\mathbf{X}^{obs}|M=1,A=a}\sigma_{a,j}^2$$

$$+\mathbb{E}_{\mathbf{X}^{obs}}(\hat{\theta} - \theta)\mathbb{E}(\mathbf{X}_{ij}^{miss}(\mathbf{X}_{ij}^{miss} - 2\mathbb{E}(\mathbf{X}_{ij}^{miss}|M_i = 1, A_i = a, \mathbf{X}^{obs})) \tag{7}$$

$$*\frac{\partial f_{imp}(\theta, M_i = 1, A_i = 1)/\partial\theta|_{\theta=\hat{\theta}}}{f_{imp}(\theta, M_i = 1, A_i = 1)}|M_i = 1, A_i = a, \mathbf{X}^{obs}) + o(\hat{\theta} - \theta)$$

Finally, by triangle inequality,

$$\mathbb{E}[\text{IFR}(f_{imp})] \geq |\mathbb{E}[\text{MSIE}_{A=0}(f_{imp})] - \mathbb{E}[\text{MSIE}_{A=1}(f_{imp})]|$$

combining equation (5) and (7) into the above expression completes the proof. $\qquad\square$

From Theorem 2, we know that the bias of the parameter estimator $\hat{\theta}$ will influence the lower bound of the imputation fairness.

## A.2 Experiment details

In this section, we provide details of the experiments in Section 4. In our experiments, FIGAN is implemented in Pytorch (version 3.6.9). MICE and MissForest are run using (`sklearn.impute.IterativeImputer`), Sinkhorn (Optimal Transport) is run using the original author's code, SoftImpute and Gain are implemented in Python. In terms of the model architecture, three fully-connected layers are adopted with Tanh activations in the generator $\mathcal{G}$, three fully-connected layers layers are adopted with ReLU activations in the dsicriminator $\mathcal{D}$. FIGAN is trained with epochs $= 50$, batch size $= 16$, learning rate $= 0.001$, $\lambda_{acc} = 1$.

For the synthetic data experiments, we choose the features at $j$-th columns with $j \in [2, 4, 7, 9, 12, 14, 17, 19]$ to have missing data, simultaneously. We generate the sensitive attribute $A$ by $A_i = 2 * \mathbf{1}\{\text{mean}(X_i) \geq 0\} - 1$. For the COMPAS dataset, we take all the 12 non-categorical features, and generate missing values on age, prior crime counts and other 4 features. After preprocessing, there are 5479 male and 1326 female recorded. For ADNI dataset, there are 643 white and 6 black patients. The influence of such significant imbalance is reflected by the values of IFR in Table 2.

