# OpenReview forum: "Fairness-aware Missing Data Imputation"
_NeurIPS.cc/2022/Workshop/TSRML — TSRML2022_

### Official Review · Reviewer_ab81 · 2022-10-11
**Intersting contribution on accounting for fairness when imputing data**

**Overall Rating:** 7

**Summary:**

The authors consider the problem of imputing data fairly.
Essentially, imputation fairness is sought by minimizing the difference in the quality of imputation between the protected groups.

They show that under certain hypotheses of the pattern for missing data (namely, when entire sub-rows of the data $X$ are systematically missing), the imputation fairness has a lower bound that is proportional to the difference between the groups in terms of the mean values for non-missing values times the variance for the imputed values.

Next, they propose a GAN-based imputation system where the generator, besides learning to reproduce the data $X$, is regularized to reduce imputation unfairness. They show that their system performs better than several baselines in terms of imputation fairness.

**Strengths:**

- The problem is interesting and this seems to be one of the first works on it (as stated by the authors and to the best of my knowledge)
- The derivations in the proof are correct
- The experimental results also seem convincing
- The paper is written very well

**Weaknesses:**

- The result of Theorem 1 (and 2) is interesting but restricted to $m_{ij} = M_{i}$. It is unclear how this result translates to the (arguably more realistic) case in which missing data is sparse. Consequently, it is unclear to me whether this result is interesting.
- GANs are notoriously hard to train. Because the experiments are a bit limited, it is not clear whether FIGAN is truly a viable approach for many different data sets, or requires a lot of trial and error to be used correctly (e.g., in terms of hyper-parameter tuning). However, I believe that the spirit of regularizing for imputation fairness is the take-home (more than the specific proposal and use of FIGAN).
- I could not find how $\lamdba_{fair}$ was chosen
- The relationship between imputation and subsequent classification/regression in terms of fairness is not elaborated upon: do we really need to impute fairly or can the use of a fair classifier overcome unfair imputation?

**Overall Recommendation:**

I think this contribution is within the scope of the workshop and I would imagine that many people in the audience will find it interesting.
IMHO, a clearer explanation of the impact of Theorem 1 and some more experiments to show that FIGAN is consistently viable, may improve the paper substantially.

**Review Confidence:**

4: The reviewer is confident but not absolutely certain that the evaluation is correct

---

### Official Review · Reviewer_Qetb · 2022-10-14
**Review for Fairness-aware Missing Data Imputation**

**Overall Rating:** 6

**Summary:**

The paper proposes a generative adversarial neural network-based approach to fair imputation of datasets containing missing values. An imputation is considered fair if the imputation error for both groups is the same. Thus, the unfairness is measured by the difference between the imputation error of the two sub-groups. The paper gives an information-theoretic lower bound on how fair an imputation algorithm can be for a given dataset. Moreover, the paper proposes a GAN-based imputer for completing the missing datasets. The objective function of the imputation method consists of three parts: optimizing the GAN parameters, optimizing the accuracy part, and optimizing the fairness regularize. Overall, the experiments demonstrate the superior performance of the proposed method in terms of accuracy and fairness on both synthetic and real datasets.

**Strengths:**

- The paper proposes an interesting and important problem. The imputation of datasets while considering sub-groups is not deeply studied to the best of my knowledge.
- Having an information-theoric lower bound on the fairness of the model reveals what we can expect ideally from a given imputation approach.
- The results are quite impressive and demonstrate a significant improvement even if we solely consider the model's accuracy.

**Weaknesses:**

- The proposed lower bound depends on the number of features that have missing values. It is a little bit confusing why it holds, even if a feature has a single missing entry. Moreover, it is not clear how the lower bound depends on the proportion of missing values on the dataset. Does it affect the variance of missing entry j in a given column? It would be nice if the authors could clarify the dependence of the lower bound on the number of samples, proportion, and type of missing values. Intuitively, it should depend on $n(1-p)/d$ where n is the number of data points, d is the dimension, and p is the proportion of missing entries.

- It seems the authors use the difference between the accuracy of two sub-groups as the fairness regularizer, which is similar to group fairness violence measures such as equalized odds (equality of opportunity,  demographic parity, or any conditional independence measure). However, to the best of my knowledge, it is impossible to directly optimize such quantities. Instead, other papers use surrogates such as Pearson Correlation [Zafar et al., 2017], Mutual Information [Donini et al., 2018], and Renyi Correlation [Mary et al., 2019]. Optimizing such quantities indirectly leads to the minimization of equalized odds violation (or demographic parity violation). It is not clear how it is possible to directly add IFR as a regularizer to the model.

- To the best of my knowledge MissForest demonstrate very impressive results when applied to datasets containing missing values. However, in the experiments, it has a very poor performance even compared to MICE. While the numbers from other packages seem totally correct based on my experience, I want to know which implementation of MissForest is used in the experiments.

- In the literature review, the authors mention two methods for inference in the presence of missing values: 1) Removing the rows containing missing values 2) Imputation of missing entries based on available data.  However, there is another type of imputation that uses distributionally robust optimization over the estimated uncertainty sets. These approaches are especially effective if the percentage of missing values is very high or the number of samples is very low. It would be nice if you mentioned this category as well. There are several sample papers under this category:
- https://www.jmlr.org/papers/volume10/xu09b/xu09b.pdf
- https://www.jmlr.org/papers/volume7/shivaswamy06a/shivaswamy06a.pdf
- https://arxiv.org/pdf/2109.00644.pdf
- https://jmlr.org/papers/volume18/17-073/17-073.pdf

- It is necessary to show the effect of hyper-parameter $\lambda_{fair}$ on the performance of the imputer. There should be a comparison between $\lambda = 0$ and the $\lambda$ chosen in the numerical results. It shows how effective the regularizer is in making the model fairer. Moreover, it is unclear whether we have a tradeoff between overall accuracy and fairness. Usually, by increasing the fairness of the model, the accuracy decreases. Is it the case here? Because it seems the model can improve both fairness and accuracy simultaneously. A comparison with another case ($\lambda = 0$) is crucial.

**Overall Recommendation:**

I recommend publishing the article if the authors address the weaknesses I mentioned. The topic is quite interesting, and it can gain attention from the fairness and robustness community.

**Review Confidence:**

5: The reviewer is absolutely certain that the evaluation is correct and very familiar with the relevant literature

---

### Official Review · Reviewer_W6Tt · 2022-10-18
**The paper is in a solid state and could spark interesting discussions**

**Overall Rating:** 6

**Summary:**

This paper considers the problem of imputation fairness, where the imputation error (MSIE) in two respective groups should be identical. A data-dependent bound on imputation fairness under optimal imputation models is derived. Furthermore, a GAN-based imputation model is presented, that places special attention on minimizing the imputation unfairness. The work is easy to follow overall and I have no major concerns regarding technical soundness.

**Strengths:**

The regularization term in the FIGAN model is constrained to make the imputation unfairness as low as possible. However, as shown in Theorem 1, there is a natural bound below which the imputation unfairness can only be decreased through degrading the imputation quality. The empirical results are satisfactory and indicate that this has not happened.

That being said, it would be a valuable sanity check to compute the bounds with optimal models and compare them to those obtained in practice. Also the trade-off between imputation fairness and imputation error could be further studied theoretically and empirically. Another question could be: How can we ensure that imputation fairness is not traded for imputation quality?


**Weaknesses:**

- I think the motivation for why precisely this notion of fairness should be useful for an end-user should be extended. What benefit does imputation fairness bring for downstream tasks? Are classifiers trained on data with higher imputation fairness also fairer with respect to conventional fairness notions, such as equal opportunity etc.?

- I would have appreciated a little more elaboration on how the hyperparameters (and which there were) for the baseline models where selected. While I assume that some hyperparameter testing was conducted, I think this essential point should be discussed further in the supplementary.

**Overall Recommendation:**

Overall, I think the paper is in a solid state and could spark interesting discussions within the workshop’s audience.


**Review Confidence:**

3: The reviewer is fairly confident that the evaluation is correct

---

### Decision · Program_Chairs · 2022-10-23

Accept